# Total Hip Arthroplasty Patients with Distinct Postoperative Fibrinolytic Phenotypes Require Different Antifibrinolytic Strategies

**DOI:** 10.3390/jcm11236897

**Published:** 2022-11-22

**Authors:** Jiacheng Liu, Bowen Chen, Xiangdong Wu, Han Wang, Xiaohai Zuo, Yiting Lei, Wei Huang

**Affiliations:** 1Orthopedic Laboratory of Chongqing Medical University, Department of Orthopedics, The First Affiliated Hospital of Chongqing Medical University, Chongqing 400016, China; 2Department of Orthopedic Surgery, Peking Union Medical College Hospital, Chinese Academy of Medical Sciences & Peking Union Medical College, Beijing 100730, China; 3Department of Orthopedics, Zhangzhou Affiliated Hospital of Fujian Medical University, Zhangzhou 363000, China

**Keywords:** total hip arthroplasty, fibrinolytic phenotype, fibrinolytic shutdown, antifibrinolysis, tranexamic acid, blood loss

## Abstract

Bleeding patients exhibit different fibrinolytic phenotypes after injury, and the universal use of tranexamic acid (TXA) is doubted. We aimed to evaluate the efficacy of postoperative antifibrinolytic treatment in total hip arthroplasty (THA) patients with different fibrinolytic phenotypes. A retrospective analysis was conducted in 238 patients who underwent THA. Patients were divided into two groups by different fibrinolytic phenotypes (non-fibrinolytic shutdown and fibrinolytic shutdown), determined by the LY30 level on postoperative day 1 (POD1). The two groups were further stratified into four sub-groups based on different postoperative TXA regimens (Group A received no TXA postoperatively, while Group B did). Hidden blood loss (HBL), decline of hemoglobin (ΔHb), D-dimer (D-D), fibrinogen/fibrin degradation product (FDP), prothrombin time (PT), activated partial thromboplastin time (APTT), and demographics were collected and compared. The clinical baseline data were comparable between the studied groups. In patients who presented non-fibrinolytic shutdown postoperatively, Group B suffered significantly lower HBL and ΔHb than Group A on POD3 and POD5. In patients who presented postoperative fibrinolytic shutdown, Group B failed to benefit from the postoperative administration of TXA when compared to Group A. No difference was found in postoperative levels of D-D, FDP, PT, and APTT. Postoperative antifibrinolytic therapy is beneficial for THA patients who presented non-fibrinolytic shutdown postoperatively, while the efficacy and necessity should be considered with caution in those with fibrinolytic shutdown. LY30 is a promising parameter to distinguish different fibrinolytic phenotypes and guide TXA administration. However, further prospective studies are needed to confirm these findings.

## 1. Introduction

Total hip arthroplasty (THA) is the most effective therapy for severe degenerative, post-traumatic, and other end-stage hip diseases [1]. However, THA is associated with overt bleeding [2]. In recent years, antifibrinolytic treatment with tranexamic acid (TXA) has been gradually brought to attention due to the excellent hemostatic efficacy among patients who underwent joint replacement [3]. Moreover, high-level evidence from multi-center randomized controlled trials (RCTs) strongly recommended the timely use of TXA in bleeding trauma patients [4,5].

However, while the fibrinolytic system is known to be activated after trauma, it does not seem to present the same phenotype all the time. Moore et al. used the percentage of clot lysis at 30 min after the clot achieved maximum strength (LY30) to define different fibrinolytic phenotypes, and found both hyperfibrinolysis (LY30 > 3%) and fibrinolytic shutdown (LY30 < 0.8%) were associated with increased mortality after injury [6]. These phenotypes may represent distinct pathophysiology and require different treatment strategies [7]. Additionally, the incidence of fibrinolytic shutdown could be up to 59%, and then researchers doubted the rationality of generalized administration of antifibrinolytic agents such as TXA in patients with fibrinolytic shutdown, since these agents could be of no benefit when there was nothing to inhibit [8,9,10]. As a result, there is a drowning realization that the individual variability in the fibrinolytic profile after injury is considerable, and maybe the administration of TXA should be limited to acute bleeding patients with non-fibrinolytic shutdown [11,12].

Postoperative administration of TXA in THA patients has been demonstrated to be beneficial in Wu et al.’s study [3]. Nevertheless, some researchers recently cautioned against the generalized use of TXA for limited blood-sparing efficacy and an increased venous thromboembolism (VTE) rate in fibrinolytic shutdown patients [13]. Therefore, in this study, we aimed to evaluate the efficacy of postoperative antifibrinolytic therapy in patients with distinct fibrinolytic phenotypes after THA.

## 2. Materials and Methods

### 2.1. Study Design

This is a retrospective study, the data of which were collected from all THA patients admitted to our center consecutively between September 2016 and November 2019. All enrolled patients received the same pre- and intra-operative antifibrinolytic treatment (1.5 g of intravenous TXA 30 min before incision and 1.0 g of topical injection before wound closure). Group A received no TXA postoperatively, while Group B was treated with 1.0 g of intravenous TXA at 3 h, 12 h, 24 h, 48 h, and 72 h after surgery. This retrospective cohort study had been approved by the ethics committee of our medical center before the data analysis. The present work was reported in line with Strengthening the Reporting of Cohort Studies in Surgery (STROCSS) [14].

### 2.2. Participants

Adult individuals who underwent primary unilateral THA were screened for eligibility. Inclusion criteria included (1) patients who received the two TXA regimens as described above; (2) hemoglobin level was over 90 g/L preoperatively; (3) negative results of preoperative doppler ultrasonography for the lower extremity. The specific exclusion criteria were as follows: (1) received bilateral hip arthroplasty; (2) received revision hip arthroplasty; (3) received tumor-related hip arthroplasty; (4) allergic to TXA; (5) combined with severe hepatic or renal failure; (6) combined with coagulation dysfunction; (7) history of venous thromboembolism within the past 6 months on admission.

### 2.3. Surgical Procedure and Perioperative Management

All THA procedures were conducted under general anesthesia by the same surgical team, consisting of three senior orthopedic surgeons. The surgical procedures were performed through the posterolateral approach with patients lying in a lateral position. A restrictive allogeneic blood transfusion (ABT) protocol (hemoglobin < 70 g/L, or the onset of symptomatic anemia with a hemoglobin level > 70 g/L) was applied in our medical center [15]. All patients received the same enhanced recovery after surgery (ERAS) procedures: (1) 1.5 g of cefuroxime sodium was administered twice daily from postoperative day 1 (POD1) to POD3 for infection prophylaxis; (2) Receiving 4000 IU of low-molecular-weight-heparin (LMWH) once daily for postoperative VTE prophylaxis. (3) Patient-controlled analgesia (PCA) pump was applied for postoperative analgesia; (4) Professional rehabilitation physicians provided one-on-one rehabilitation treatment postoperatively; (5) Dietitians provided individualized dietary nutrition treatment; (6) Clinical pharmacists guide and adjust the perioperative drug therapy to achieve the stability of combined underlying diseases.

### 2.4. Outcome Measurements

The primary outcomes were hidden blood loss (HBL) and decline of hemoglobin (ΔHb). HBL was calculated based on the perioperative levels of hematocrit (Hct) through the formula reported by Gross and Nadler [16,17]. ΔHb was acquired directly through the differences of hemoglobin levels on different days.

The secondary outcomes included D-dimer (D-D), fibrinogen/fibrin degradation product (FDP), prothrombin time (PT), and activated partial thromboplastin time (APTT). Blood samples were routinely tested on admission and POD1, POD3, POD5, and POD7. Demographic characteristics, including gender, age, height, weight, body mass index (BMI), primary diagnosis, and operation time, were also recorded and compared.

### 2.5. Thromboelastography (TEG)

Conventional coagulation measures could provide limited value in arthroplasty, as there has been a persistent state of hypercoagulability after arthroplasty, while the standard coagulation tests would reflect normal parameters and suggest a balanced coagulation [18,19,20]. Of note, TEG is an ideal assessment technique that could reveal more details towards the hemostatic process and coagulability changes, and thus could be utilized to detect the potential coagulation and fibrinolysis changes during TXA use [21,22,23].

TEG includes seven parameters: (1) Reaction time ®, period to 2 mm amplitude, represents enzymatic reaction function; (2) Kinetics (K), period from 2 to 20 mm amplitude, represents clot kinetics; (3) Alpha angle (α-Angle), slope between R and K, represents fibrinogen level; (4) Maximum amplitude (MA), represents maximum platelet function; (5) Percentage of clot lysis at 30 min after MA (LY30), represents fibrinolytic activity; (6) Estimate percent lysis within 30 min after MA (EPL), represents fibrinolytic activity; (7) Comprehensive coagulation index (CI), represents a linear combination of R, K, α-Angle and MA values [23]. Enrolled patients were divided into different fibrinolytic phenotypes by the tested level of LY30 on POD1: (1) Non-fibrinolytic shutdown (Hyper- and physiologic fibrinolysis), LY30 ≥ 0.8%; (2) Fibrinolytic shutdown, LY30 < 0.8%. The TEGs collected in this study were performed by TEG^®^ Hemostasis Analyzer, Model 5000 (Haemonetics Corporation, Braintree, MA, USA).

### 2.6. Statistical Analysis

Study data were collected and managed by using Excel (Microsoft Corporation, Washington, DC, USA), while data analyses were conducted by SPSS version 24.0 (IBM Corporation, Armonk, NY, USA). Descriptive analyses of the participants included reporting means (with SDs) for continuous variables and frequencies (with percentages) for categorical or discrete variables. For continuous outcomes, the independent *t*-test was used to analyze independent normal distributed variables, while the Wilcoxon Mann-Whitney U test was performed for non-normal distribution variables. Qualitative variables were analyzed through the chi-square test. All tests were two-sided with statistical significance set at a *p*-value < 0.05.

## 3. Results

### 3.1. Patient Demographics

A total of 271 patients that underwent THA during the study period were screened, and 238 were eligible for this study (Figure 1). Enrolled patients with different postoperative fibrinolytic phenotypes were divided into two groups (Figure 1). This study aimed to evaluate the efficacy of postoperative antifibrinolytic therapy in THA patients who presented distinct fibrinolytic phenotypes postoperatively. Therefore, the two groups were further stratified into four sub-groups based on the different postoperative TXA regimens (received postoperative TXA or not). Demographics and baseline characteristics of the patients are listed in Table 1. No significant difference was found between the two groups with respect to gender, BMI, primary diagnosis, or other baseline data.

### 3.2. Primary Outcomes

The dynamic changes of HBL and ΔHb among the studied groups were shown in Figure 2, and the detailed values were listed in Table 2. For patients who presented non-fibrinolytic shutdown postoperatively, Group B suffered significantly less HBL than Group A on POD3 and POD5 (590.05 ± 227.15 mL vs. 768.10 ± 335.10 mL, *p* = 0.009; 520.88 ± 227.88 mL vs. 769.34 ± 313.01 mL, *p* = 0.003). Consistently, the ΔHb were significantly lower in Group B on POD3 and POD5 when compared with Group A (21.83 ± 10.92 g/L vs. 27.42 ± 10.48 g/L, *p* = 0.027; 20.23 ± 10.19 g/L vs. 26.25 ± 9.80 g/L, *p* = 0.033). However, for the patients with fibrinolytic shutdown, there was no significant difference in HBL and ΔHb between the groups.

### 3.3. Secondary Outcomes

Detailed information regarding postoperative D-D, FDP, PT, and APTT was given in Table 3. Despite different postoperative fibrinolytic phenotypes, no difference was found in the four mentioned parameters on POD1, POD3, POD5, and POD7.

## 4. Discussion

The main findings obtained in this study showed that: (1) The fibrinolytic system appeared to present different phenotypes following the THA procedure, and fibrinolytic shutdown accounted for up to 63.03%. (2) When considering blood conservation, THA patients with evidence of non-fibrinolytic shutdown benefited from the postoperative sequential use of TXA, while those with fibrinolytic shutdown did not. (3) The LY30 level tested on POD1 may help to distinguish different fibrinolytic phenotypes and determine whether the postoperative antifibrinolytic treatment is needed.

A coagulation cascade is initiated by the injury of the vascular bed and the exposure of tissue factor, and hemostasis is later achieved with the formation of stable clots on the site of the vascular injury [24,25]. TXA could competitively block the lysine binding site on plasminogen to impede its activation, and then inhibit the activity of the fibrinolytic system [26]. Therefore, TXA decreases bleeding secondary to fibrinolysis by stabilizing the clots that seal the broken vessels [27]. Generalized use of TXA was recommended in a large-scale RCT (CRASH-2) involving 20,211 bleeding trauma patients, as the mortality was significantly reduced in those receiving TXA after injury [4]. Similarly, another multi-center RCT (WOMAN) also found decreased death due to post-partum hemorrhage in the TXA-group, and suggested the timely administration of TXA soon after bleeding onset [28]. Timing seems to be a critical factor that influences the effect of TXA. Later in an exploratory analysis of the results from CRASH-2, researchers also concluded that TXA should be given soon after injury in bleeding trauma patients, as no benefit was found in delayed use [29]. However, as the study of fibrinolysis after trauma is deepened, timing does not seem to be the only factor that have influence on the effect of TXA.

It is known that fibrinolysis is a physiological process, which occurs parallel with coagulation and plays the role of a “brake” to prevent the extension of clot formation beyond the focus of injury [24]. When this process is upregulated pathologically, it is termed hyperfibrinolysis. Chakrabarti et al. first reported the biphasic fibrinolytic response after injury, and defined the concept of fibrinolytic shutdown [30]. Fibrinolytic shutdown is an inhibited state of fibrinolysis at the opposite spectrum from hyperfibrinolysis. Recently, TXA was only recommended in patients with solid evidence of non-fibrinolytic shutdown, and researchers cautioned against the use of TXA in patients with fibrinolytic shutdown [9,12]. This is in consistent with our results in this study that elective THA patients presented non-fibrinolytic shutdown postoperatively suffered significantly less blood loss by receiving multiple doses of TXA, while those with fibrinolytic shutdown did not. Notably, it may help to explain this phenomenon that TXA improved the fibrin clot strength in patients with hyperfibrinolysis, but failed to strengthen that with fibrinolytic shutdown [31]. Therefore, antifibrinolytic therapy seems to be unreasonable in patients with fibrinolytic shutdown, since there is in fact nothing to inhibit.

Use of TXA on an as-needed basis is of vital importance because patients with fibrinolytic shutdown are concerned to be at risk of VTE if antifibrinolytic agents are administered empirically [13]. D-D is one of the most widely used parameters to assess fibrinolytic activity. However, D-D seems to be a less reliable parameter to guide the administration of TXA. Patients with elevated D-D levels and low TEG measured fibrinolytic activity were speculated to be at a state of occult hyperfibrinolysis [13]. One explanation for this paradoxical phenomenon is that the fibrinolytic system has previously activated and has been shut down by the time of blood draw [31]. The D-D level remains high due to the long residence time in the circulation, while the fibrinolytic function is actually already shut down. Our findings are in consistent with this hypothesis, as we found the D-D levels in the studied groups were similar on POD1, POD3, POD5, and POD7. Therefore, LY30 may be a more reliable parameter to guide the administration of TXA than D-D, as it could assess fibrinolytic activity in real-time.

There are several limitations of our study. First, although we applied strict inclusion/exclusion criteria to control the potential bias, this retrospective study is limited by the non-randomized design, which means the results could still be influenced by the heterogeneity of patients. Second, the findings of this study were concluded based on our single medical center, which means there may be some inevitable representative bias. Third, the blood loss was calculated based on the Hb and Hct levels tested on different days after surgery, which may easily be influenced by postoperative rehydration treatment. However, the perioperative rehydration protocol was the same in our center, and thus there should be no difference in the comparison between groups. Last but the least, due to the retrospective study design, we failed to provide favorable follow-up data, especially ultrasonography examination to evaluate postoperative VTE rate.

## 5. Conclusions

Overall, the results obtained in our study suggested that not all the patients receiving THA need postoperative antifibrinolytic treatment. Patients with evidence of non-fibrinolytic shutdown benefited from the administration of TXA after surgery, while those with fibrinolytic shutdown did not. The LY30 level acquired on POD1 could help to distinguish different fibrinolytic phenotypes and guide a more accurate and appropriate administration of TXA. However, these findings need to be demonstrated by future prospective studies.

## Figures and Tables

**Figure 1 jcm-11-06897-f001:**
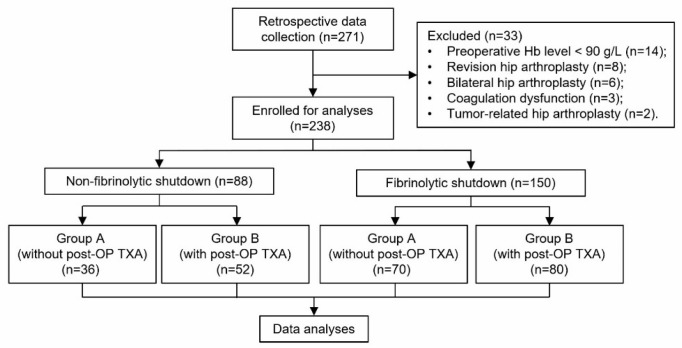
The flow chart of study enrolment. Hb, hemoglobin; post-OP, postoperative; TXA, tranexamic acid.

**Figure 2 jcm-11-06897-f002:**
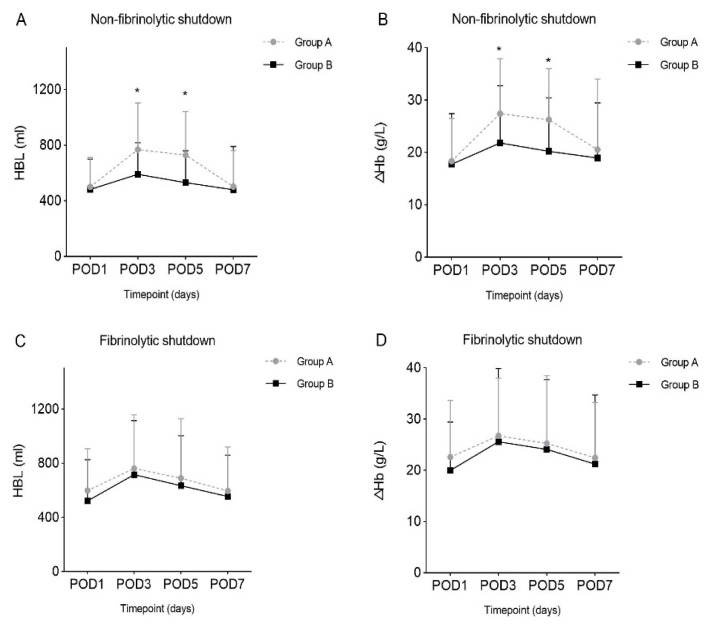
Dynamic changes of postoperative hidden blood loss and decline of hemoglobin levels. (**A**) In patients who presented non-fibrinolytic shutdown, postoperative hidden blood loss between Groups A and B. (**B**) In patients who presented non-fibrinolytic shutdown, decline of hemoglobin levels between Groups A and B. (**C**) In patients who presented fibrinolytic shutdown, postoperative hidden blood loss between Groups A and B. (**D**) In patients who presented fibrinolytic shutdown, decline of hemoglobin levels between Groups A and B. HBL, hidden blood loss; POD, postoperative day; ΔHb, decline of hemoglobin levels; *, *p* < 0.05.

**Table 1 jcm-11-06897-t001:** Comparison of demographics by study groups.

Demographics	Non-Fibrinolytic Shutdown (n = 88)			Fibrinolytic Shutdown (n = 150)		
	Group A (n = 36)	Group B (n = 52)	*p* Value	Group A (n = 70)	Group B (n = 80)	*p* Value
Female, n (%)	21 (58.3%)	30 (57.7%)	0.952 ┼	38 (54.3%)	48 (60.0%)	0.480 ┼
Age, year (SD)	60.38 (14.70)	63.12 (17.88)	0.096 *	68.03 (15.67)	68.72 (14.80)	0.793 *
Height, cm (SD)	159.38 (6.55)	160.00 (7.47)	0.726 *	157.59 (6.12)	159.65 (7.81)	0.150 *
Weight, kg (SD)	58.08 (8.09)	59.79 (9.36)	0.434 *	59.43 (9.56)	57.75 (11.54)	0.423 *
BMI, kg/m^2^ (SD)	22.89 (3.17)	23.23 (3.29)	0.667 †	23.65 (3.17)	22.59 (3.87)	0.143 *
Major diagnosis						
HF, n (%)	12 (33.3%)	13 (25.0%)	0.473 ┼	25 (35.7%)	23 (28.8%)	0.362 ┼
HOA, n (%)	4 (11.1%)	8 (15.4%)	0.754 ┼	10 (14.3%)	15 (18.8%)	0.464 ┼
DDH, n (%)	9 (25.0%)	12 (23.1%)	0.835 ┼	18 (25.7%)	19 (23.8%)	0.781 ┼
ONFH, n (%)	11 (30.6%)	19 (36.5%)	0.560 ┼	17 (24.3%)	23 (28.8%)	0.537 ┼
Left THA, n (%)	22 (61.1%)	31 (59.6%)	0.888 ┼	31 (44.3%)	41 (51.3%)	0.394 ┼
Intraoperative blood loss, mL (SD)	138.70 (99.90)	165.19 (156.42)	0.141 *	132.46 (117.45)	157.39 (135.37)	0.319 *
Operation time, min (SD)	81.50 (33.81)	84.79 (33.13)	0.676 †	96.93 (42.00)	86.00 (36.80)	0.122 *
LOS, day (SD)	13.51 (5.45)	14.02 (4.22)	0.560 *	13.26 (3.10)	13.96 (3.88)	0.315 *

┼ Chi-square test; * Independent-samples *t*-test; † Mann-Whitney U-test; BMI, body mass index; HF, hip fracture; HOA, hip osteoarthritis; DDH, developmental dysplasia of hip; ONFH, osteonecrosis of femoral head; THA, total hip arthroplasty; LOS, length of stay.

**Table 2 jcm-11-06897-t002:** Post-operative hidden blood loss and decline of hemoglobin level.

Outcomes	Non-Fibrinolytic Shutdown (n = 88)			Fibrinolytic Shutdown (n = 150)		
	Group A (n = 36)	Group B (n = 52)	*p* Value	Group A (n = 70)	Group B (n = 80)	*p* Value
HBL, mL (SD)						
POD1	500.91 (211.17)	481.46 (218.04)	0.713 †	597.92 (308.45)	522.23 (304.26)	0.230 *
POD3	768.10 (335.10)	590.05 (227.15)	0.009 *,§	761.87 (396.77)	715.93 (398.98)	0.572 †
POD5	729.34 (313.01)	520.88 (227.88)	0.003 *,§	689.53 (439.38)	634.21 (368.88)	0.275 *
POD7	501.48 (259.81)	479.86 (310.61)	0.864 *	595.10 (325.65)	553.80 (306.59)	0.700 *
ΔHb, g/L (SD)						
POD1	18.36 (8.16)	17.75 (9.65)	0.769 *	22.57 (11.05)	19.98 (9.45)	0.167 *
POD3	27.42 (10.48)	21.83 (10.92)	0.027 †,§	26.75 (11.24)	25.59 (14.30)	0.623 *
POD5	26.25 (9.80)	20.23 (10.19)	0.033 †,§	25.25 (13.21)	24.08 (13.65)	0.666 †
POD7	20.55 (13.46)	18.94 (10.51)	0.723 *	22.46 (10.79)	21.20 (13.50)	0.767 *

† Mann-Whitney U-test; * Independent-samples *t*-test; § Statistically significant difference in mean values between groups; TXA, tranexamic acid; HBL, hidden blood loss; POD, postoperative day; ΔHb, decline of hemoglobin level.

**Table 3 jcm-11-06897-t003:** Post-operative levels of four common coagulation and fibrinolysis parameters.

Outcomes	Non-Fibrinolytic Shutdown (n = 88)			Fibrinolytic Shutdown (n = 150)		
	Group A (n = 36)	Group B (n = 52)	*p* Value	Group A (n = 70)	Group B (n = 80)	*p* Value
D-D, mg/L (SD)						
POD1	4.50 (1.99)	4.08 (4.04)	0.743 *	5.57 (3.24)	3.80 (3.65)	0.052 *
POD3	2.30 (1.16)	1.64 (1.43)	0.164 *	2.03 (1.57)	1.60 (1.36)	0.252 *
POD5	6.82 (7.33)	5.59 (5.39)	0.283 *	4.38 (2.19)	3.76 (2.06)	0.328 †
POD7	4.30 (0.19)	3.94 (2.50)	0.846 *	5.80 (3.61)	3.84 (1.85)	0.174 *
FDP, μg/mL (SD)						
POD1	12.34 (3.52)	10.08 (7.26)	0.324 *	13.15 (8.37)	10.44 (8.81)	0.222 *
POD3	7.60 (2.76)	6.27 (6.41)	0.508 *	6.91 (3.50)	6.39 (5.40)	0.699 *
POD5	10.80 (7.38)	10.07 (3.38)	0.801 *	12.13 (4.89)	11.38 (5.24)	0.607 †
POD7	12.00 (5.24)	11.40 (1.56)	0.876 *	14.70 (9.76)	11.86 (5.07)	0.465 *
PT, s (SD)						
POD1	13.73 (1.21)	13.78 (0.84)	0.872 *	13.75 (0.65)	13.88 (0.89)	0.558 *
POD3	14.25 (2.14)	13.86 (1.06)	0.382 *	13.79 (0.99)	13.76 (0.95)	0.923 †
POD5	13.60 (2.19)	13.72 (0.81)	0.794 *	13.21 (0.94)	13.46 (1.04)	0.413 *
POD7	14.15 (0.64)	13.90 (1.19)	0.776 *	12.25 (0.21)	13.77 (1.34)	0.122 *
APTT, s (SD)						
POD1	36.83 (5.51)	36.78 (4.89)	0.978 *	36.33 (5.10)	38.31 (5.38)	0.147 †
POD3	41.23 (6.33)	41.70 (7.68)	0.850 *	40.77 (5.48)	43.72 (7.33)	0.133 *
POD5	40.24 (5.72)	41.49 (6.20)	0.621 *	38.85 (4.89)	41.55 (6.60)	0.140 *
POD7	42.85 (8.41)	40.16 (5.16)	0.507 *	34.25 (4.03)	40.77 (5.02)	0.082 *

* Independent-samples *t*-test; † Mann-Whitney U-test; D-D, D-dimer; POD, postoperative day; FDP, fibrinogen/fibrin degradation product; PT, prothrombin time; APTT, activated partial thromboplastin time.

## Data Availability

The data will be available from the corresponding author on reasonable request.

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
