# Peer review of "Total Hip Arthroplasty Patients with Distinct Postoperative Fibrinolytic Phenotypes Require Different Antifibrinolytic Strategies"

_jcm, 2022, doi:10.3390/jcm11236897_

Round 1

Reviewer 1 Report

1. This retrospective study is limited by  the non-randomized design, especially postoperative (Group A, no TXA), (Group B, TXA) (lines 69 - 70).

2. TEG® Hemostasis Analyzer is one of the most sensitive and specificity? (lines 125 - 126)

3. Is there a connection between TXA and fibrinolytic shutdown? (lines 209 - 213)

4. The use of TXA in patients with fibrinolytic shutdown increased the mortality?

5. Please explain why LY30 remain a more reliable parameter to guide the administration of TXA? (lines 224 - 225)

Reviewer 2 Report

Thank you for an interesting paper.  The possibility that a significant portion of our patients may not benefit from, and may even be harmed by, TXA administration is an important one.  I would be curious to know whether fibrinolytic shutdown occurs at the time of insult (surgery), or at what time period post-operatively, as this may have implications for timing of post-op TXA administration.

Some of your post-op care is what I would consider non-standard (72h of TXA administration, 3 days of post-op antibiotic coverage).  In particular, I would like to see you comment on whether the pre- and intra-op doses of TXA are useful even in the fibrinolytic group.

I would also like to know how accessible and expensive the TEG/LY30 testing is, and whether you recorded any DVT/PE in any patients.
